# Health workers' perspectives on school-based mass drug administration control programs for soil-transmitted helminthiasis and schistosomiasis in Ogun State, Nigeria

Folahanmi T. Akinsolu[1,2☯]*, Olunike R. Abodunrin[1,3☯], Mobolaji T. Olagunju[4☯], Ifeoluwa E. Adewole[1‡], Oluwabukola M. Ola[1‡], Chukwuemeka Abel[1‡], Rukayat Sanni-Adeniyi[1‡], Nurudeen O. Rahman[5☯], Olukunmi O. Akanni[1‡], Diana W. Njuguna[6☯], Islamiat Y. Soneye[7‡], Abideen O. Salako[1,2☯]*, Oliver C. Ezechi[1,2☯]*, Orsolya E. Varga[8☯], Olaoluwa P. Akinwale[1,2☯]

1 Department of Public Health, Lead City University, Ibadan, Oyo State, Nigeria, 2 Clinical Sciences Department, Nigerian Institute of Medical Research, Lagos, Lagos State, Nigeria, 3 Department of Planning and Research, Lagos State Health Management Agency, Lagos, Lagos State, Nigeria, 4 Department of Epidemiology and Biostatistics, Nanjing Medical University, Nanjing, China, 5 Department of Medical Parasitology and Infection Biology, Swiss Tropical and Public Health Institute, Basel, Switzerland, 6 School of Nursing, Dedan Kimathi University of Technology, Nyeri, Kenya, 7 Department of Public Health, Ogun State Ministry of Health, Ota, Ogun State, Nigeria, 8 Department of Public Health and Epidemiology, University of Debrecen, Debrecen, Hungary

☯ These authors contributed equally to this work.
‡ These authors also contributed equally to this work
* Folahanmi.tomiwa@gmail.com (FTA); salako.abideennaheem@gmail.com (AOS); oezechi@yahoo.co.uk (OCE)

**Data Availability Statement:** All relevant data are within the manuscript.

## Abstract

### Background

Soil-transmitted helminthiasis (STH) and schistosomiasis (SCH) are among the most prevalent neglected tropical diseases (NTDs), affecting 1.5 billion globally, with a significant burden in sub-Saharan Africa, particularly Nigeria. These diseases impair health and contribute to socio-economic challenges, especially in children, undermining educational and future economic prospects. The 2030 NTD Roadmap highlights Mass Drug Administration (MDA) as a critical strategy for controlling these NTDs, targeting vulnerable populations like school-age children. Despite some successes, challenges persist, indicating the need for deeper insights into program implementation. This study focuses on the perspectives of health workers implementing MDA in selected local government areas (LGAs) of Ogun State, Nigeria, aiming to identify challenges and enablers that align with the broader NTD 2030 goals.

### Methodology/Principal findings

The study used a qualitative research approach involving focus group discussions and in-depth interviews with health workers engaged in neglected tropical disease control programs in Ogun State, Nigeria, between July and September 2022. A semi-structured

**Funding:** This work received financial support from the United States Agency for International Development (USAID) through its Neglected Tropical Diseases Program of through their support of the Coalition for Operational Research on Neglected Tropical Diseases (COR-NTD) grant to FTA. COR-NTD is funded at The Task Force primarily by the Bill & Melinda Gates Foundation and USAID. The funders had no role in study design, data collection and analysis, decision to publish, or preparation of the manuscript.

**Competing interests:** The authors have declared that no competing interests exist.

questionnaire guided the exploration of ideas, and the data were analyzed using the QRS Nvivo 12 software package. The study found that the school-based MDA control program's efficacy largely relies on strong collaborations and partnerships, particularly with educators, community heads, and other stakeholders. These alliances and strategic communication methods, like town announcements and media campaigns, have been pivotal in reaching communities. However, the program does grapple with hurdles such as parental misconceptions, limited funds, insufficient staffing, and misalignment with the Ministry of Education. It is recommended to boost funding, foster early stakeholder involvement, enhance mobilization techniques, and consider introducing a monitoring card system similar to immunization.

## Conclusions/Significance

The MDA Integrated Control Programs for STH and SCH in Ogun State schools demonstrate a holistic approach, integrating knowledge, collaboration, communication, and feedback. Health workers have shown commitment and adeptness in their roles. However, achieving maximum efficacy requires addressing critical barriers, such as parental misconceptions and funding challenges. Adopting the recommended strategies, including proactive communication, increased remuneration, and introducing a tracking system, can significantly enhance the program's reach and impact. The involvement of all stakeholders, from health workers to community leaders and parents, is essential for the program's sustainability and success.

## Introduction

Soil-transmitted helminthiasis (STH) and schistosomiasis (SCH) rank among the top neglected tropical diseases (NTDs) globally, affecting around 1.5 billion people, primarily in areas with limited resources [1,2]; approximately 600 million of this global burden originate in Africa [1,3,4]. Nigeria has the highest prevalence of NTDs in Sub-Saharan Africa [5,6]. Thus, an alarming 20 million individuals yearly require treatment for SCH [7]. These parasitic infections disproportionately affect vulnerable populations, especially children, and perpetuate a cycle of poverty, impaired cognitive development, and reduced school attendance [6–8].

The 2030 NTD Roadmap has a strategic focus on controlling and eliminating NTDs, which has a significant portion of its success in implementing integrated mass drug administration (MDA) programs [9,10] as the primary approach for controlling and potentially eliminating NTDs. MDA involves periodically administering safe, effective, low-cost drugs to at-risk populations until disease-specific targets are met [11].

This initiative commonly entails the distribution of specific medications, such as deworming pills or preventive treatments for NTDs like SCH and STH, to substantial cohorts of children within schools and communities where these infections are endemic [12,13].

As recommended by the WHO, a single dose of albendazole (400 mg) or mebendazole (500 mg) in MDA to treat STH is efficient [10,14]. Notable, the MDA program initiative has played a pivotal role in alleviating the burden of SCH and STH [15,16]. In an endemic area, the WHO recommends that at least 75% of school-age children (SAC) should receive MDA [17,18]. This approach helps control and combat NTDs effectively. However, certain studies have also indicated that despite the implementation of multiple rounds of MDA targeting STH and SCH in

SAC, there has been no significant change in the prevalence of this disease [13,19,20]; thus, the barriers to MDA program might affect the attainment of the 2030 global NTD goals [3,13].

The effectiveness of MDA programs for STH and SCH is not without challenges; it can be influenced by many factors, including the engagement and perspectives of frontline health workers [2]. As the crucial link between the healthcare system and the community, health workers play a crucial role in successfully implementing MDA programs. Health workers facilitate MDA by distributing medication, educating communities, and monitoring progress to control and eliminate diseases effectively. Therefore, the health worker's perceptions will provide insights into the challenges and enablers of the MDA control programs and significantly improve the programs, thus aiding in achieving the NTD 2030 goals [21–23].

The objectives of this study were to identify the challenges faced by health workers during MDA implementation in selected local government areas (LGAs) in Ogun State and explore the enablers that can contribute to successful MDA program execution.

## Methodology

### Ethical statement

The study was conducted following the principles outlined in the Declaration of Helsinki, and the study participants' data were treated with strict confidentiality throughout the study. The study received ethical approval from the Department of Health Planning, Research, and Statistics, Ogun State Ministry of Health, Ogun State, Nigeria (reference number HPRS/351/388, approval date June 16, 2021). Verbal informed consent was obtained from all study participants after the objective and purpose of the study were explained. Informed assent forms were provided to children and minors and informed parental consent was obtained from their parents or local guardians. For illiterate participants, impartial witnesses were enlisted to assist with the consenting process. Participants were included in the study only after signing the informed consent/assent form and receiving a paper copy.

### Study settings

Ogun State, Nigeria, was selected as the study location due to its high prevalence of STH (19.2%) and SCH (32.2%) and the inconsistent support of partner programs for NTD control [24,25]. MDA commenced in Ogun State 22 years ago directed to STH and SCH treatment [25]. Four LGAs in Ogun State were purposively selected based on recommendations from the Department of Public Health, Ogun State Ministry of Health. The following LGAs were included in the study: Ikenne, with a prevalence of 12.6% for STH and 1.46% for SCH; Abeokuta North, with a prevalence of 22.4% for STH and 15.40% for SCH; Abeokuta South, with a prevalence of 27.5% for STH and 1.61% for SCH; and Obafemi Owode, with a prevalence of 21.1% for STH and 8.78% for SCH [24].

### Study design

A qualitative research approach was chosen for this study to delve deeply into the nuanced aspects of the MDA program. This approach was selected because it allows for a rich, detailed understanding of the complex realities and experiences of healthcare workers, educators, and other stakeholders involved in NTD control programs in Ogun State, Nigeria. Qualitative methods are particularly effective in capturing the diverse perspectives, attitudes, and social dynamics that quantitative methods might overlook, which is crucial for uncovering the underlying factors that influence the effectiveness of program implementation and addressing the barriers to success [26,27]. Thematic analysis was employed to extract key themes from the

data collected which was instrumental in providing a comprehensive understanding of the challenges faced by NTD control programs in addressing STH and SCH. It offered valuable insights into the barriers to achieving effective control interventions and highlighted the importance of understanding the context-specific factors that impact the success of health initiatives. The study's timeline, from July 2021 to September 2021, allowed for an in-depth exploration during a critical period for these health programs.

## Study participants

The study focused on stakeholders in the fight against STH and SCH in areas where these diseases are endemic. Participants were selected for key informant interviews (KIIs) and focus group discussions (FGDs) using purposive sampling based on their roles within the health system, community positions, and involvement in managing NTDs. In-depth KIIs were conducted with the Director of Public Health (DPH), Ogun State Ministry of Health, Medical Officers of Health (MoHs), and Local NTD officers (LNTDs). Additionally, FGDs were held with community health extension workers (CHEWs), who serve as health educators (See Fig 1). A semi-structured questionnaire guided the exploration of ideas during these sessions.

## Sampling technique

A purposive sampling technique was employed to select a diverse cohort of health workers, each actively engaged in the management of STH and SCH. This approach was designed to capture a comprehensive and targeted array of perspectives surrounding the control and prevention of STH and SCH.

Participants were chosen based on several well-defined criteria, which include their roles within the health system were primarily considered, focusing on those with direct responsibilities related to the management, treatment, and prevention of STH and SCH, ensuring that the study encapsulated a variety of viewpoints from different tiers of the healthcare system, providing a multifaceted understanding of the challenges and strategies in managing these diseases.

The study further narrowed potential participants to those with explicit involvement and experience managing NTDs. These individuals were part of the planning, execution, or evaluation phases of control and prevention programs, providing insights grounded in practical experience.

Geographically, the health workers were selected from LGAs that were endemic with STH and SCH. This criterion was vital to ensure that participants had firsthand experience and understanding of the diseases within the specific context of the local environment and population dynamics.

The study also focused on participants' positions and expertise, aiming for a representative mix of roles. The DoPH from the Ogun State Ministry of Health was included to offer a policy and administrative perspective. The MoHs and LNTDs were chosen for their direct involvement in local health programs and specialized focus on NTDs, respectively. This selection provided insights into the practical and intricate challenges of managing NTDs at various levels. The CHEWs, renowned for their role as health educators and close interaction with the community, were selected for FGDs to provide a grassroots perspective on community response, educational challenges, and potential enablers for improving MDA programs.

Lastly, the duration of experience with NTD programs was considered. The study aimed to include a mix of veterans with years of experience and newer members with fresh perspectives, ensuring a rich and varied collection of insights.

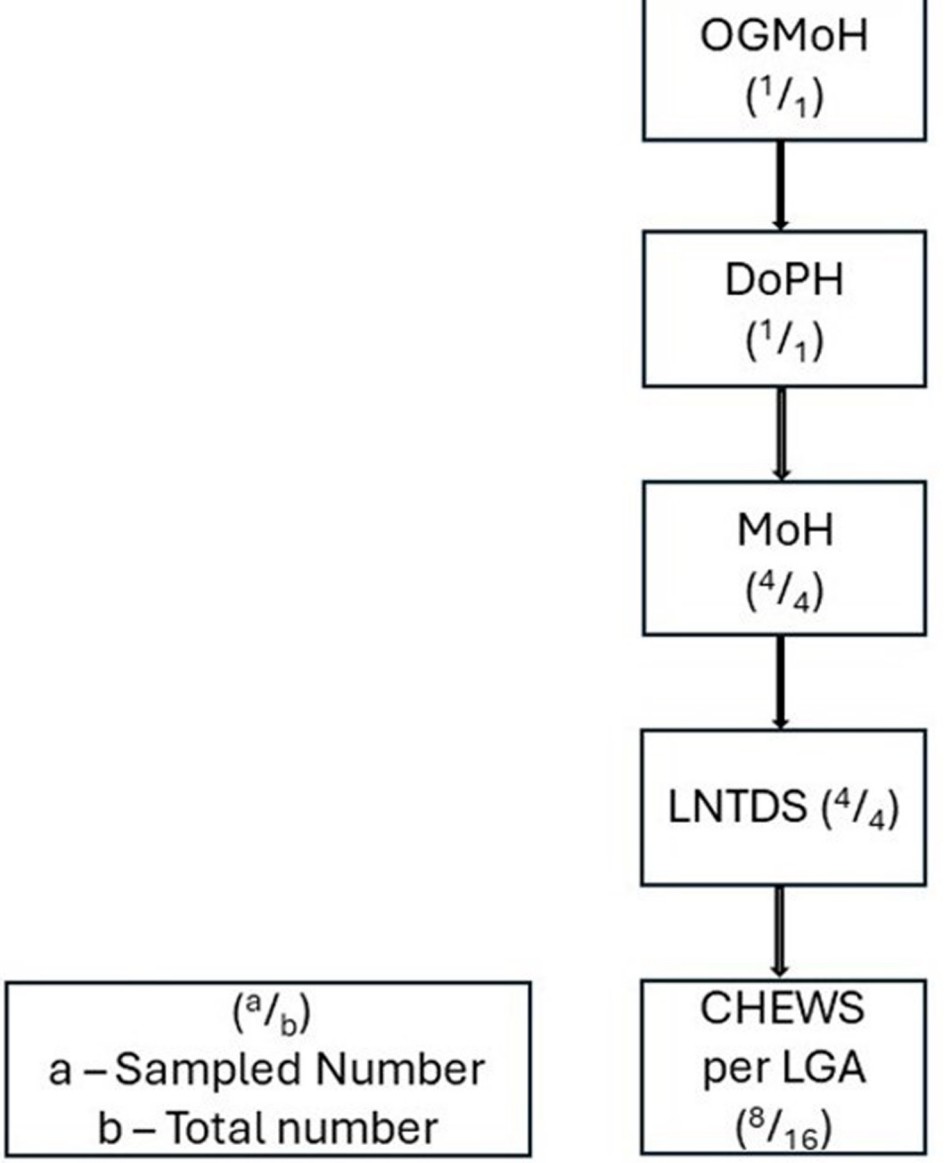

**Fig 1. Flow chart of stakeholder's selection.**

### Sample size determination

In this qualitative study, sample size calculation was not applicable. Instead, the number of FGDs and KIIs was determined by the concept of saturation. Saturation was when no new information or themes were observed in the data [28]. To ensure a comprehensive exploration of the topic and to account for the depth and complexity of the subject matter, a certain number of FGDs and KIIs were initially planned for.

A total of 4 FGDs and 9 KIIs were conducted. The criteria for reaching saturation included the repetition of key themes, the emergence of a coherent pattern in the data, and the judgment that additional interviews would likely not yield new insights [28]. This judgment was made by assessing the data after each session and determining if further sessions were

necessary. The decision to stop collecting data was made when the information became redundant, and the research team was confident that the collected data provided a rich and thorough understanding of the study topic [29].

By relying on these criteria for saturation, the study ensured that the data collected were sufficient to develop a well-rounded and in-depth understanding of the health workers' perspectives on MDA program implementation without prescribing a rigid number of interviews from the outset.

## Study tools

The study utilized two primary tools for data collection: an IDI guide for community health workers, health educators, and opinion leaders and a KII guide for MDA managers (See Table 1).

These guides were carefully developed to ensure a comprehensive exploration of the perspectives and experiences of participants regarding the MDA programs for STH and SCH.

The IDI and KII guides were pre-tested to ensure the questions' clarity, relevance, and appropriateness. The pre-testing was conducted with a small group of individuals with characteristics like those of the study population who were not part of the main study. This process helped refine the questions, ensuring they were understandable and elicited the information needed. Adjustments were made to the wording and order of questions based on feedback from the pre-test. For instance, complex medical jargon was simplified, and ambiguous questions were clarified to ensure they were comprehensible to participants with varying levels of education and expertise.

Several measures were taken to ensure the reliability of the study tools. These measures include refining the questions after the pilot study to ensure clarity and the solicitation of the required information, validating findings with participants to accurately reflect their experiences accurately, and engaging in peer debriefing. All interviewers underwent rigorous

**Table 1. Key elements of the interview guide.**

| Interview Guide | | Key Elements |
|---|---|---|
| **FGDs** | 1 | Community Illnesses & NTDs |
| | 2 | Prevalence and Vulnerable Groups |
| | 3 | Preventive Measures & Treatment |
| | 4 | Community Efforts & Drug Availability |
| | 5 | Program Awareness & Involvement |
| | 6 | Communication & Barriers |
| | 7 | Program Reception & Rumors |
| | | |
| **IDI/KIIs** | 1 | Health Priorities & NTD Importance |
| | 2 | NTD Occurrence & MDA Program Awareness |
| | 3 | Communication & Barriers |
| | 4 | Program Reception & Rumors |
| | 5 | Data Management & Legislation |
| | 6 | Involvement & Health Sector Priorities |
| | 7 | Perceptions of NTD Programs |
| | 8 | Integration with Health Programs |
| | 9 | Program Evaluation & Stakeholder Involvement |
| | 10 | Policies & Suggestions |

training to understand the study's objectives, familiarize themselves with the tools, and conduct interviews consistently. The training emphasized the importance of understanding 'good' qualitative research practices, including the ethical dimensions of conducting sensitive and impactful research. The training introduced the research assistants to various qualitative data collection methods, such as IDIs and FGDs, each integral to comprehending the complex tapestry of human experiences. A significant portion of the training was devoted to ethical considerations, ensuring that the research assistants understood their role in conducting research with the utmost respect, dignity, and responsibility. Also, the training delved into ethnographic fieldwork, teaching research assistants to observe, participate, record, and reflect effectively. Role-playing exercises and field visit simulations were integral, sharpening their practical skills and preparing them for real-world complexities. Feedback sessions were incorporated to foster reflection and enhance their interviewing and fieldwork techniques.

Following the initial pre-test, a pilot test was conducted in a setting similar to the study site to test the tool and assess the interviewers' performance and ability to maintain neutrality and consistency. Throughout the data collection period, the research team held regular debriefing sessions to discuss challenges, share experiences, and ensure a common understanding and approach to administering the tools. Transcripts from the audio recordings were regularly checked against the original recordings for accuracy. The research team discussed and resolved discrepancies to ensure the data accurately represented the participants' statements. A systematic thematic analysis was conducted to identify and analyze patterns within the data. This approach helped to ensure that the findings were grounded in the data and that the analysis was reliable and valid.

## Data collection

Between July and September 2021, we conducted 4 FGDs with CHEWs and 9 KIIs with DoPH, MoHs, and LNTDs involved in the school-based MDA control program. The semi-structured and FGDs interview topics were developed and pre-tested to ensure the data's reliability. Audio tape recorders captured all the information obtained during the KIIs and FGDs. The interviewers also made observations and took notes while recording to ensure accuracy. The interviews were conducted in both English (the national language of Nigeria) and Yoruba (the local language of the study participants) in various settings, including small halls, private rooms, and offices. The duration of each KII ranged from 30–40 minutes, and the FGDs took 35–45 minutes. FGDs were conducted in the participants' respective LGAs.

## Data processing and analysis

Thematic analysis was employed to extract key themes from the data collected. This qualitative technique was instrumental in providing a comprehensive understanding of the challenges faced by NTD control programs in addressing STH and SCH. It offered valuable insights into the barriers to achieving effective control interventions and highlighted the importance of understanding the context-specific factors that impact the success of health initiatives. Data from FGDs and KIIs were transcribed, coded, and analyzed thematically based on the emerging themes of knowledge, collaboration and partnership, communication, perceived barriers, and recommendations. The recorded audio in the Yoruba language was transcribed into English. Transcription was done independently among the team members and then checked for verification and accuracy with simultaneous audio playing. The study content was analyzed thematically, indexed, and coded inductively using the QRS NVivo 12 software package (QRS International, Doncaster, Australia) for the content analysis of unstructured qualitative data. The initial open codes were sorted into sub-themes based on their similarity. QRS NVivo 12

software played a pivotal role in the data analysis process of the study, serving as a tool for managing, organizing, and analyzing the unstructured qualitative data collected from FGDs and KIIs. According to the participant's responses, these sub-themes were clustered and refined to form broad themes and debated within the research team. Data collection, analysis, and reporting followed the Standards for Reporting Qualitative Research (SPQR) guidelines.

Guba and Lincoln's criteria for determining rigor in qualitative research were used to ensure consistency during protocol preparation, data collection, development of a coding system, interrater reliability, and data analysis [30]. Two interviews per group were coded by two authors (FT & OR), for which the degree of similarity was determined by calculating the interrater reliability using the QRS NVivo 12 software. To ensure the rigor and reliability of the thematic analysis, the research team employed a collaborative approach to resolve any discrepancies that arose during the coding process. This approach was facilitated through several steps. Initially, FT and OR independently coded a subset of the data to apply the understanding of the coding framework without each other influence. After the independent coding phase, FT and OR convened to compare and discuss the coding, which involved a line-by-line review of the coded data, where each discrepancy was discussed in detail with other team members, MT and DW. The purpose of this discussion was not only to resolve the specific discrepancy but also to refine the collective understanding of the coding framework. When discrepancies arose, the team worked together to reach a consensus, which involved revisiting the original data and the definitions within the coding framework. An audit trail was maintained throughout the process, documenting decisions and their rationale, including notes from the discussion and any coding framework or code definition changes. The audit trail is a valuable resource for understanding how discrepancies were resolved and provides transparency in the analytic process. Periodically, inter-coder reliability checks were conducted to quantify the level of agreement. These checks served as a quality control measure, ensuring the codes were applied consistently and reliably over time. Cohen proposed that Kappa scores should be interpreted using the following: values $\leq 0$ indicate lack of agreement, values 0.01–0.20 indicate no or little agreement, values 0.21–0.40 indicate fair agreement, values 0.41–0.60 indicate moderate agreement, values 0.61–0.80 indicate substantial agreement, and values 0.81–1.00 indicate nearly perfect agreement. Kappa's score for the thematic analysis was 1.00, indicating nearly perfect agreement [31].

## Results

### Composition of key informant interview and focused group discussion participants

A total of 41 health workers were interviewed. 4 FGDs were conducted amongst the CHEW, while 9 KIIs were conducted among the DoPH, MoHs, and LNTDs. (See Table 2) Table 3 shows the socio-demographic characteristics of the study participants in the KIIs and FGDs.

The study findings were categorized into five thematic areas: Knowledge, Collaboration and Partnerships, Communication, Perceived Barriers, and Recommendations, as shown in

**Table 2. Composition of key informant interview and focused group discussion.**

| Participants | Female | Male | Total |
|---|---|---|---|
| CHEW | 23 | 9 | 32 |
| LNTD | 4 | - | 4 |
| MOH | 2 | 2 | 4 |
| DoPH | - | 1 | 1 |

**Table 3. Socio-demographic characteristics of the study participants in the KIIs and FGDs.**

| Characteristics | | CHEW N (%) | LNTD N (%) | MoH N (%) | DoPH N (%) |
|---|---|---|---|---|---|
| **Gender** | | | | | |
| | Male | 23 (71.9) | 0 (0.0) | 2 (50.0) | 1 (100.0) |
| | Female | 9 (28.1) | 4 (100.0) | 2 (50.0) | 0 (0.0) |
| **Age in years** | | | | | |
| | 30–35 | 14 (43.8) | 0 (0.0) | 0 (0.0) | 0 (0.0) |
| | 36–40 | 5 (15.6) | 1 (25.0) | 0 (0.0) | 0 (0.0) |
| | 41–45 | 12 (37.5) | 3 (75.0) | 1 (25.0) | 0 (0.0) |
| | 46–50 | 1 (3.1) | 0 (0.0) | 2 (50.0) | 1 (100.0) |
| | 51–55 | 0 (0.0) | 0 (0.0) | 1 (25.0) | 0 (0.0) |
| **Marital Status** | | | | | |
| | Single | 4 (12.5) | 0 (0.0) | 0 (0.0) | 0 (0.0) |
| | Married | 21 (65.6) | 4 (100.0) | 3 (75.0) | 1 (100.0) |
| | Divorced | 5 (15.6) | 0 (0.0) | 1 (25.0) | 0 (0.0) |
| | Widowed | 2 (6.3) | 0 (0.0) | 0 (0.0) | 0 (0.0) |
| **Religion** | | | | | |
| | Christianity | 19 (59.4) | 3 (75.0) | 2 (50.0) | 1 (100.0) |
| | Islam | 13 (40.6) | 1 (25.0) | 2 (50.0) | 0 (0.0) |
| **Years of experience** | | | | | |
| | 3–5 | 12 (37.5) | 2 (50.0) | 0 (0.0) | 0 (0.0) |
| | 6–8 | 20 (62.5) | 2 (50.0) | 0 (0.0) | 0 (0.0) |
| | 9–11 | 0 (0.0) | 0 (0.0) | 1 (25.0) | 0 (0.0) |
| | 12–15 | 0 (0.0) | 0 (0.0) | 3 (75.0) | 1 (100.0) |

CHEW: Community Health Extension Workers.

LNTD: Local Neglected Tropical Diseases officers.

MOH: Medical Officers of Health.

DoPH: Director of Public Health.

Table 4, ensuring an in-depth exploration of factors influencing program efficacy and formulating informed recommendations. The participants were health workers from the Ogun State Ministry of Health whose socio-demographic characteristics demonstrated the appropriateness of each category. From the health workers' perspective, the study participants described key components of MDA-integrated control programs for STH and SCH.

## Theme 1: Knowledge

All the health workers interviewed had adequate knowledge and were involved in the school-based MDA Integrated Control Programmes for STH and SCH.

## Sub-theme 1: Knowledge of the MDA program

When asked if the participants were familiar with the MDA program, an MoH from one of the LGAs said,

*"I know MDA well, and there has been a positive response from the recipients."*

Another LNTD officer who has participated in the MDA program said that,

**Table 4. Thematic areas.**

| Themes | Sub-themes | Challenges | Enablers |
|---|---|---|---|
| Knowledge | Knowledge of the MDA program<br>Knowledge of the disease | NA | Involvement in the pre-and post-planning stages |
| Collaboration and Partnerships | Support | Poor involvement of the Ministry of Education | Training of teachers |
| | Potential partners identification | | |
| Communication | Strategy of communication | incentives | Existing structure |
| | Medium of communication | | Involvement of grassroots contacts |
| Perceived Barriers | | | |
| | Staffing issues: funding, movement of drugs (logistically) | NA | NA |
| | Insufficient support from MOE | | |
| Recommendations | Technical recommendation: proper planning and involvement of stakeholder | NA | NA |
| | Support- financial | | |

NA: Not Applicable.

*"Yes, I am very conversant with it because yesterday we did one at a school, and I monitored them."*

## Sub-theme 2: Knowledge of the disease

Healthcare workers' knowledge of STH and SCH is pivotal, influencing MDA program success through accurate diagnosis, treatment, and community education for effective control and prevention.

*"Yes! A worker needs to know STH and SCH. Even when treating them, we do get feedback that some of the children are vomiting worms, which means if they didn't take that drug, it would have caused another thing in their system."*

The CHEW officers stated that,

*"Uhmmmm! The health centers' statistics show they are endemic diseases because children cannot do without playing in soil and water."*

## Enabler

In pre- and post-planning stages, involvement is pivotal to health workers' knowledge of the MDA program. When health workers are aware of the project in reasonable detail and involved in the stages of planning before execution and after, they will have good knowledge about what is obtainable on the project.

The CHEW educates and immunizes both children and adults in the community, said,

*"I am familiar with MDA and recently administered drugs to the communities."*

Another CHEW shared her involvement in the program. She stated;

*"I have been involved in the pre-implementation and post-implementation stage. During the pre-implementation, we identified forms in the school and community. The stakeholders call parents who don't want their children to take it, and we call religious leaders to meetings and solicit their assistance in the community. Then, during post-implementation, we want to see how many children have accessed the drugs, what points we want to clap on ourselves that we have done well, and what areas of challenges we like to identify to avoid in our next round."*

### Theme 2: Collaboration and Partnerships

Collaboration and partnerships are pivotal for the MDA program's success, fostering shared expertise, resources, and diverse perspectives to address complex health challenges and maximize positive outcomes effectively.

### Sub-theme 1: Technical support

Effective collaboration and partnerships, with robust support mechanisms, are indispensable for healthcare workers to succeed in mass drug administration programs, ensuring comprehensive coverage and sustainable impact. Technical support is needed and essential, which was evident when the health workers were asked if they collaborated or partnered with other people on the MDA control programs for STH and SCH. A CHEW stated,

*"We work with some teachers and train them because the children do not want to accept the drugs."*

Most health workers opined that engaging community leaders fosters trust and increases community compliance rates.

An LNTD officer cited a situation in which the community leader was rescued. He stated that,

*"Some of them do resist it, but when we call the community leader, he also helps us to explain to them, and they cooperate with us."*

An LNTD officer stated that,

*"Community leaders help us disseminate the information, and we, as LNTD officers, go to them again when we want to distribute letters, we go again to tell them."*

### Sub-theme 2: Financial support

Financial support, a critical facet of collaboration and partnership in mass drug administration programs among healthcare workers, enhances resource allocation, training, and infrastructure, ultimately optimizing program efficacy and sustainability.

A CHEW elaborated on their partnership with the school and how their joint effort has helped them in the success of the MDA program, stating that,

*"Some schools invite us to PTA meetings where issues are addressed at the PTA meetings. After the meeting, drug administrations started immediately. The staff and educators collaborate in Abeokuta South."*

This type of collaboration usually involves money in some ways. The absence of this would imply that the training or collaborative effort would suffer for it.

## Enablers

The availability of training programs and platforms for teachers on the MDA program is a potent enabler that ensures that collaborating teachers are brought to speed and intimated on the program while the parent runs the idea at a meeting such as the PTA meeting.

## Theme 3: Communication

Health workers acknowledged the different communication strategies that create a communication synergy that contributes to the effectiveness of the MDA Integrated Control Programmes for STH and SCH.

## Sub-theme 1: Strategy of communication

Strategic communication within mass drug administration programs among healthcare workers is paramount, ensuring effective dissemination of information, fostering understanding, and promoting community engagement for successful implementation.

One of the MoHs stated that,

*"There's good communication between Local Government and the community. There's a process to which it will be done; we go through common head leaders, and through Town announcers, he will announce it to convince them of acceptability."*

An LNTD officer also stated that,

*"We communicate the school-based program through communication mobilization and sensitization (radio jingles, flyers, posters, and outdoor van campaign), which is very effective."*

From the focus group discussion among the CHEW, a situation that occurred a few years ago was stated;

*"A female student slept after taking the drugs till closing time, and the next day, we were told she was dead."*

Also, they mentioned that,

*"The parents still find a way to say don't give my child."*

The situation proffers the need to communicate with parents. Hence, she stated,

*"We have the parents' numbers to get their consent for their child to be given or not; this is because of some of the issues encountered before."*

Conversely, the synergy in the communication is not only with the community but also with stakeholders, sponsors, and the State Government.

### Sub-theme 2: Medium of communication

Selecting an appropriate communication medium is vital in mass drug administration programs among healthcare workers, optimizing message delivery, and ensuring widespread comprehension for successful implementation and community participation.

A CHEW stated that,

*"Our communication with sponsors is okay; there are various WhatsApp groups where we interact."*

### Enablers

**Existing structures:** The MDA program has been structured to open lines of communication with academic stakeholders in different levels of education as affected by the program, ensuring that the newly registered institutions stand a high chance of being incorporated into the program as much as existing institutions.

**Involvement of grassroots contact:** eligible students are ensured to be covered in whichever setting they are in.

As stated by one of the MoHs on the education sector's role in the program,

*"Both public and private schools ensure all eligible students are being covered, and this process has been very effective."*

This opens the communication line between the principals or heads of the schools, which in turn gets to the parents of the students.

### Theme 4: Perceived Barriers

The study evaluated perceived barriers among health workers regarding School-Based MDA Integrated Control Programmes for STH and SCH.

The CHEW pointed out that,

*"Parents are the challenge. Some are adamant. So many false beliefs (it kills, it's for sterilization.)"*

Another CHEW confirmed the point mentioned above. She stated that,

*"Well! According to my job, I can say Ignorance is the problem. Some people rely on what happened in the past, and the influence of people on each other affects the program."*

An LTND officer opined that,

*"False information like a taboo; two children died! If we can eradicate such, the program will go smoothly."*

### Sub-theme 1: Staffing Issues-Funding, movement of drugs (Logistics)

Health workers noted that their small remuneration and inadequate staffing affect their efficiency,

A CHEW opined that,

*"Health workers alone cannot do the work alone because we are understaffed. Workers are not being paid, and there are insufficient mobilizers to work."*

*"Two Community Distributors cannot cover a whole community. They take drugs to their houses, which is stressful to operate."*

While another CHEW stated that,

*"The stipend attached to mobilization is two thousand Nigerian naira ($2.65), which is small. In this present-day Nigeria, it is not good."*

An LNTD officer opined that,

*"Funding is a shortcoming for this program, so we couldn't go to each school and class to inform the children, and also we can't reach the out-of-school children."*

A MoH also shared what he perceived as the barrier.

*"All we need more is being able to mobilize people and sensitization; that is what is not optimal."*

### Sub-theme 2: Insufficient support of MOE

Another MoH noted the shortcomings of the Ministry of Education.

*"Poor participation of the Ministry of Education. The school is not as coordinated as expected because the schools often don't know about the program. They will say they have not been told that some people are coming, so they are trying to know whether the school is registered."*

### Theme 5: Recommendation

The health workers shared different suggestions to improve the School-Based MDA Integrated Control Programmes for STH and SCH.

### Sub-theme 1: Support-financial

Most of the CHEW suggested more mobilization and increasing their remuneration.

*"More hands to be on deck if there's a way we can involve people in the community to ensure more coverage and mobilizers."*

*"The issue of funding, if possible, they should increase the funding. Ensure more community participation through the mobilizers."*

*"Government should increase their fund to increase their work rate, and also Government should provide a means mobilizing."*

Similarly, one LNTD officer opined that,

*"If they can fund the program very well, I think it will encourage those people doing it, then the mobilization program also; jingles and posters."*

A MoH shared the same thought,

*"Funding is key; increase funding so that more schools and teachers are involved. Like this last one, we have a school where only a teacher was involved, the same teacher will write, it can't work if there is no fund let's not do it at all because in the school at least we need to train more than one let's have at least three or four you understand then have more schools, involve the private schools, there are kids there too. By then, more people will be captured, giving us data of how many people and all that.'*

A CHEW also recommended incentives for partners,

*"Partners that help work within the community should be given a token. They should be given identification tags and materials or aprons. The drugs can be printed on the aprons. It can be retrieved, wasted, and reused. Posters tear easily from pain and sweat."*

### Sub-theme 2: Technical recommendation: proper planning and involvement of stakeholder

A MoH also noted the importance of proper planning and the involvement of stakeholders.

*"All parties involved will make the program sustainable. "*

*"Involve parents, especially mothers, in the infant welfare clinic,"*

Another MoH stated that,

*"Preparation ahead of time will stop the crash of the program."*

In addition, he opined that,

*"People should be involved at the planning stage."*

Some health workers recommended a more proactive approach that can be considered. An LNTD officer stated that,

*"NTD messages can be brought in health education, just like vaccination, family planning, etc."*

A MoH opined that,

*"If it can be like immunization, which is done when there is a card, you understand, that makes it smooth so the child knows when to get it so the facility they have it rather than wait for the programs. Let everybody know that they have a card to follow because many times you get to the school, the kids may not even remember. For example, we had last year but could not see if it was October, November, or December. So likewise for the kids, when last did you take, you say, last year? But if there was a card, everybody could follow up; I know when I am due. The program should just run like we have routine immunization. That is the policy I feel will be nice for those kids 5–14."*

## Discussion

Evaluating the understanding of health workers engaged in school-based MDA control programs regarding the challenges and facilitators of the program is crucial for its success [32] to ensure that comprehensive knowledge and necessary actions are aligned to achieve program objectives across all levels. According to Piotrowski et al. and Kabatereine et al., it's essential to consider insights from experienced health workers, the frontline individuals who directly interact with the target population in the program's initial stages [32,33]. Our study showed that health workers possess adequate knowledge of the program, demonstrating experience in their roles. Furthermore, it was evident that their engagement provided them with valuable opportunities for active community-based participation.

Collaboration is integral to the success of the MDA program [34–36]. Project managers need to partner with stakeholders who can enhance the program's effectiveness. In this study, the collaborative efforts showcased by health workers, schools, and community leaders underscore the significance of partnerships in achieving success in MDA programs targeting STH and SCH. As highlighted by the CHEW, the involvement of teachers in the MDA program demonstrates a strategic approach to overcoming reluctance among children. This collaboration improves the acceptance of drugs and ensures that children receive essential treatment for these NTDs. Molyneaux et al. identified collaboration as essential for strengthening the health system [37]. The NTD agenda extends beyond parasitic diseases, with a massive annual delivery of medicines to a billion people, mainly targeting the poorest, aligning with the 2030 UN SDGs for universal health coverage; adapting to global events and policy changes is crucial for sustained progress, requiring innovative thinking within health systems, led by policymakers, managers, and frontline health workers, which necessitates effective collaboration among partners involved [3,11,18,38]. Such partnerships between health workers and educators showcase the potential for leveraging existing networks within communities to achieve program goals. Partnerships also enhance community engagement, build trust, address resistance, and streamline program implementation [2,39]. Acknowledging and nurturing such collaborations will be crucial to reducing the burden of these NTDs and improving the health and well-being of affected communities. These findings agreed with the study conducted in Cambodia on collaboration and engagement in the MDA control programs [35].

Effective collaboration in MDA programs targeting STH and SCH faces several challenges. Divergent interests and goals among stakeholders, such as health workers prioritizing disease control and educators focusing on academic continuity, can lead to conflicts [13]. Additionally, resource constraints, encompassing financial, human, and logistical aspects, pose significant obstacles to sustained collaboration. Limited resources may result in competition among stakeholders, hindering their ability to work together effectively [40,41]. Moreover, communication barriers, including differences in professional language and inadequate infrastructure, further complicate collaboration efforts by impeding the smooth flow of information and coordination [35,40]. Addressing these challenges necessitates a comprehensive approach that considers diverse stakeholder perspectives, allocates resources judiciously, and establishes clear communication channels to foster collaboration in MDA programs (27) [13,32].

Effective communication among stakeholders and involved parties is critical to an MDA control program's success [13]. Haiti's NTD control program achieved a national scale driven by strong communication and good epidemiological coverage in MDA. This success is owed to crucial elements, including resource communication, logistical coordination, and a community awareness campaign [41]. Our study highlighted how healthcare workers recognize the

synergistic enhancement of MDA control programs for STH and SCH through diverse communication strategies. Health workers acknowledged effective communication with community leaders and stakeholders, facilitating program acceptance and local support for field officers, and this aligns with findings in other programs [36]. Findings from studies indicated that integrating technology and innovative communication methods can serve as valuable tools to improve the effectiveness of MDA programs for STH and SCH [35,38]. This enhancement encompasses various aspects, including heightened awareness, knowledge dissemination, acceptance, demand generation, behavior change communication (BCC), feedback mechanisms (such as monitoring cards), quality assurance (ensuring drug availability), accountability (through reporting), and sustainability (facilitated by stakeholder engagement) [38,42,43]. Technology and innovative communication methods play pivotal roles in augmenting the effectiveness of MDA control programs targeting STH and SCH [13]. The success of Haiti's NTD control program on a national scale underscores the significance of diverse communication strategies, including resource communication, logistical coordination, and community awareness campaigns [41]. Therefore, incorporating technology and innovative communication contributes significantly to the success and impact of MDA programs for STH and SCH by addressing multiple facets of program implementation and community engagement [22,32,44].

The study's findings shed light on perceived barriers among health workers in implementing School-Based MDA control programs for STH and SCH. These barriers were identified by health workers closely engaged with local communities. They highlighted that parents' resistance to the program poses a significant challenge. This resistance often stems from false beliefs, including the misconception that the treatment is lethal or is aimed at sterilization. Our findings highlighted similar STH and SCH misperceptions from studies in Nigeria, Bangladesh, and the Philippines [2,21,22,45]. These misconceptions can impede parents' willingness to allow their children to participate and undermine the success of health programs. However, it underscores the importance of effective community engagement and communication strategies. Addressing these misconceptions through targeted awareness campaigns and involving influential community members can foster a better understanding of the program's intentions and benefits.

Another barrier to the program effectiveness the study identified was insufficient staffing and low remuneration. Health workers stressed that their workload is exacerbated by inadequate numbers, hindering their ability to reach their target population effectively. Moreover, the limited remuneration to mobilizers undermines their motivation and capacity to engage in the program entirely. Hence, a common barrier highlighted was the inadequate funding allocated to the program. This shortage of resources limits the program's scope, preventing comprehensive coverage and outreach efforts, as stated in similar studies [19,22,46,47].

This study focused on health workers' perspectives regarding the challenges and facilitators of school-based MDA control programs for SCH and SCH in Ogun State, Nigeria. The study highlighted the understanding and active community engagement of health workers' insights as crucial for program success. Collaboration emerges as essential, showcased by health workers, educators, and community leaders working together to overcome challenges. Effective communication is critical in enhancing MDA control program integration and community support. The study identified barriers such as parental misconceptions and resource limitations, suggesting awareness campaigns and improved incentives as solutions.

We recommended a comprehensive approach to enhance school-based MDA control programs and strategies for controlling STH and SCH, which involves tailored community engagement involving local leaders to dispel misconceptions and ensure accurate information dissemination [25,47]. Collaboration among health workers, educators, and community

leaders, supported by effective communication strategies, is crucial for program acceptance. Addressing workforce challenges, advocating for funding diversity, proper planning, and prioritizing capacity-building are essential for achieving lasting success. Implementing the recommended comprehensive approach for enhancing school-based MDA control programs for STH and SCH may encounter several challenges. Tailored community engagement, while effective, may face resistance due to cultural nuances or entrenched beliefs. Collaboration among health workers, educators, and community leaders may be challenging, requiring concerted efforts to align diverse perspectives and agendas. Limited resources or technological barriers might hinder effective communication strategies in certain regions. Addressing workforce challenges and advocating for funding diversity may face bureaucratic hurdles, and unforeseen contextual factors might impede proper planning. Prioritizing capacity-building could be resource-intensive and time-consuming. Successful implementation of these recommendations necessitates overcoming these challenges through adaptable and context-specific approaches, ongoing assessment, and a commitment to addressing emerging obstacles to ensure sustained and impactful outcomes.

While the recommended comprehensive approach for enhancing school-based MDA control programs for STH and SCH is promising, its implementation has potential drawbacks and challenges. Although effective, tailored community engagement may encounter resistance rooted in cultural nuances or entrenched beliefs, posing a barrier to disseminating accurate information. Achieving collaboration among health workers, educators, and community leaders may prove challenging, requiring concerted efforts to align diverse perspectives and agendas, potentially slowing down the implementation process. Effective communication strategies, vital for program acceptance, could face hindrances in regions with limited resources or technological barriers, impacting the reach and impact of information dissemination. Addressing workforce challenges and advocating for funding diversity may encounter bureaucratic hurdles, potentially delaying crucial support for the program. Proper planning might be impeded by unforeseen contextual factors, requiring adaptive strategies. Additionally, while essential, prioritizing capacity-building could be resource-intensive and time-consuming, posing a challenge to swift and widespread implementation. Overcoming these drawbacks demands context-specific approaches, continuous assessment, and a commitment to addressing emerging obstacles to ensure sustained and impactful outcomes from the proposed recommendations.

The study conducted in Ogun State, Nigeria, provides valuable insights into the challenges and enablers of school-based MDA control programs for addressing NTDs. However, several limitations exist: translation from Yoruba to English might introduce biases; there's potential for self-reporting bias from participants; thematic analysis is inherently subjective; reliance on audio recordings might miss non-verbal cues; data collector training does not guarantee consistent data quality; and cultural norms of the region might influence responses in a way not applicable elsewhere.

## Conclusion

The insights of health workers engaged in school-based MDA control programs for STH and SCH are paramount for optimizing its success. The health workers' firsthand experience provides invaluable feedback regarding the challenges faced and the solutions that have proven effective. This research highlights the crucial importance of collaboration not only within healthcare professionals but also between them, educators, and community leaders. These alliances improve community acceptance and streamline the effective execution of MDA control initiatives. Successful communication serves as a cornerstone in dispelling misconceptions

and gaining community backing. Despite challenges like parental misconceptions, limited resources, staffing issues, and inadequate compensation, these hurdles can be overcome through targeted community involvement, awareness initiatives, and appropriate incentivization.

## Acknowledgments

We extend our deepest gratitude to all the individuals and institutions that contributed to the success of this study. First, we would like to sincerely thank the health workers, community leaders, and participants from Ogun State, Nigeria, who generously shared their time, experiences, and insights. Their valuable contributions were instrumental in shaping the research and understanding the complexities of managing Soil-transmitted helminthiasis and schistosomiasis mass drug administration control programs.

We are also grateful to the Department of Public Health, Faculty of Basic Medical and Health Sciences, Lead City University, Ibadan, Nigeria; Nigerian Institute of Medical Research, Lagos State; and the University of Debrecen, Hungary, for providing the necessary facilities and support for this research.

## Author Contributions

**Conceptualization:** Folahanmi T. Akinsolu, Diana W. Njuguna, Abideen O. Salako, Oliver C. Ezechi, Orsolya E. Varga, Olaoluwa P. Akinwale.

**Data curation:** Folahanmi T. Akinsolu, Mobolaji T. Olagunju, Diana W. Njuguna, Oliver C. Ezechi, Orsolya E. Varga, Olaoluwa P. Akinwale.

**Formal analysis:** Folahanmi T. Akinsolu, Olunike R. Abodunrin, Mobolaji T. Olagunju, Ifeoluwa E. Adewole, Oluwabukola M. Ola, Nurudeen O. Rahman, Diana W. Njuguna, Abideen O. Salako, Orsolya E. Varga, Olaoluwa P. Akinwale.

**Investigation:** Folahanmi T. Akinsolu, Olunike R. Abodunrin, Mobolaji T. Olagunju, Ifeoluwa E. Adewole, Rukayat Sanni-Adeniyi, Olukunmi O. Akanni, Diana W. Njuguna, Olaoluwa P. Akinwale.

**Methodology:** Folahanmi T. Akinsolu, Olunike R. Abodunrin, Mobolaji T. Olagunju, Ifeoluwa E. Adewole, Oluwabukola M. Ola, Chukwuemeka Abel, Rukayat Sanni-Adeniyi, Nurudeen O. Rahman, Olukunmi O. Akanni, Diana W. Njuguna, Islamiat Y. Soneye, Oliver C. Ezechi, Orsolya E. Varga, Olaoluwa P. Akinwale.

**Project administration:** Olunike R. Abodunrin, Oluwabukola M. Ola.

**Supervision:** Folahanmi T. Akinsolu, Olunike R. Abodunrin, Nurudeen O. Rahman, Olukunmi O. Akanni, Diana W. Njuguna, Oliver C. Ezechi, Orsolya E. Varga, Olaoluwa P. Akinwale.

**Validation:** Olunike R. Abodunrin, Nurudeen O. Rahman, Olukunmi O. Akanni.

**Visualization:** Mobolaji T. Olagunju.

**Writing – original draft:** Folahanmi T. Akinsolu, Olunike R. Abodunrin, Mobolaji T. Olagunju, Ifeoluwa E. Adewole, Oluwabukola M. Ola, Chukwuemeka Abel, Rukayat Sanni-Adeniyi, Nurudeen O. Rahman, Diana W. Njuguna, Islamiat Y. Soneye, Abideen O. Salako, Oliver C. Ezechi, Orsolya E. Varga, Olaoluwa P. Akinwale.

**Writing – review & editing:** Folahanmi T. Akinsolu, Olunike R. Abodunrin, Mobolaji T. Olagunju, Ifeoluwa E. Adewole, Oluwabukola M. Ola, Chukwuemeka Abel,

Rukayat Sanni-Adeniyi, Nurudeen O. Rahman, Olukunmi O. Akanni, Diana W. Njuguna, Islamiat Y. Soneye, Abideen O. Salako, Oliver C. Ezechi, Orsolya E. Varga, Olaoluwa P. Akinwale.

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
