## [Decision Letter · Decision Letter 0]

15 Dec 2023

PONE-D-23-31424Health Workers' Perspectives on School-Based Mass Drug Administration Control Programs for Soil-Transmitted Helminths and Schistosomiasis in Ogun State, Nigeria.PLOS ONE

Dear Dr. Akinsolu,

Thank you for submitting your manuscript to PLOS ONE. After careful consideration, we feel that it has merit but does not fully meet PLOS ONE’s publication criteria as it currently stands. Therefore, we invite you to submit a revised version of the manuscript that addresses the points raised by the two reviewers.

We look forward to receiving your revised manuscript.

Kind regards,

Hammed Oladeji Mogaji, Ph.D

Academic Editor

PLOS ONE

Journal Requirements:

2. In the ethics statement in the Methods, you have specified that verbal consent was obtained. Please provide additional details regarding how this consent was documented and witnessed, and state whether this was approved by the IRB

"This work received financial support from the United States Agency for International

Development (USAID) through its Neglected Tropical Diseases Program of through their support 

of the Coalition for Operational Research on Neglected Tropical Diseases (COR-NTD) grant to 

FTA. COR-NTD is funded at The Task Force primarily by the Bill & Melinda Gates Foundation

and USAID. The funders had no role in study design, data collection and analysis, decision to

publish, or preparation of the manuscript"

"The author(s) received no specific funding for this work"

Reviewers' comments:

Reviewer's Responses to Questions

**Comments to the Author**

1. Is the manuscript technically sound, and do the data support the conclusions?

Reviewer #1: Yes

Reviewer #2: Partly

2. Has the statistical analysis been performed appropriately and rigorously? 

Reviewer #1: N/A

Reviewer #2: No

3. Have the authors made all data underlying the findings in their manuscript fully available?

Reviewer #1: Yes

Reviewer #2: Yes

4. Is the manuscript presented in an intelligible fashion and written in standard English?

Reviewer #1: Yes

Reviewer #2: Yes

5. Review Comments to the Author

Reviewer #1: The manuscript reports the results of a study that explored health worker perspectives on School-based MDA implementation in Ogun's selected LGAs, pinpointing challenges and enablers so as to enhance the effectiveness of the program and align with the NTD 2030 goal of elimination. Data was collected using qualitative methods i.e. Key Informant Interviews and Focus Group Discussions amongst various cadres of health workers who participate in the implementation of the School-based deworming for STH and SCH.

This is a very important topic in the area of SCH and STH elimination.

Comments:

Paginate the document for ease of reference.

Line 184-186 states that 9 KIIs were conducted whilst on line 136, it is indicated that 11 KIIs were conducted. Which one is the correct position? Please address for consistency.

Results section: I suggest that the authors tease out the sub-themes under each of the outlined themes i.e. Knowledge, Collaboration and Partnership, Communication, Perceived Barriers and Recommendations. This will help in the flow of presentation. For example Theme 4. Perceived Barriers has several sub-themes: 1) inadequate knowledge about MDA which should be reported together with inadequate mobilization and sensitization to avoid fragmentation in presentation; 2) staffing issues which include understaffing and poor remuneration; 3) insufficient support from MOE…..

What do the results say about knowledge of STH and SCH because adequate information on etiology of these infections is important for MDA uptake?

Table 1: Thematic areas need more details- Themes on column one, sub-themes on column two, challenges on column three and enablers on column four. These should then be presented in narrative form.

Line 125-127 on study design and participants, it is indicated that KIIs were conducted with LNTDs officers, MOHs, DPH and health educators but the health educators are missing on line 184-186 and on table 2 and also none of the presented quotes is from the health educators. Please address this discrepancy.

Line 195 cannot start with “another LTND” as none other is mentioned on lines 191-194.

For the CHEWs FGDs, the presentation should be revised to reflect the views of the group (s) and not of the individual CHEW.

Line 396, limitation on generalizability of the findings is not necessary since qualitative studies are not meant to be generalized but rather for theory building and informing the design of quantitative studies.

Line 398 on sample size is also not a limitation and thus not necessary. Qualitative studies by design do not have sample sizes and number of study participants is based on saturation point since the information being sought is on individual feelings, perceptions, beliefs etc. and is therefore very subjective.

Line 401 on dominant voice overshadowing others, this should also be removed as a limitation as it should have been managed during data collection if the FGDs were conducted in adherence to standard procedures.

Conclusion section could be summarized and avoid expounding much of what has already been presented in the discussion section e.g. sentence on lines 412 to 414.

Reviewer #2: Elaborate on the rationale for selecting the qualitative research approach. Clarify the criteria used for determining saturation in the absence of a formal sample size calculation. Elaborate on the process of pretesting interview topics and how reliability was ensured. Provide additional details on the content covered during the three-day training for data collectors. Consider providing more context on how discrepancies in coding were resolved among the research team. Clearly outline the selection criteria for health workers participating in the study. Provide additional details on the content of the semi-structured questionnaire. Include more information on the demographic characteristics of the study participants. Clarify the role of QRS NVivo 12 software in data analysis.

6. PLOS authors have the option to publish the peer review history of their article (what does this mean?). If published, this will include your full peer review and any attached files.

Reviewer #1: **Yes: **Dr Doris Njomo

Reviewer #2: No

---

## [Author Response · Author response to Decision Letter 0]

13 Jan 2024

Dear Reviewer I,

I would like to sincerely thank the reviewers for their insightful comments and suggestions regarding our manuscript titled "Health Workers' Perspectives on School-Based Mass Drug Administration Control Programs for Soil-Transmitted Helminthiasis and Schistosomiasis in Ogun State, Nigeria." We have carefully considered each comment and have made corresponding revisions to the manuscript. Below, we address each comment in detail and outline the changes we have made to improve our work.

Specific Comments:

1. Title: Revise the title to accurately reflect the disease, using "Soil-Transmitted Helminthiasis" or "Soil-Transmitted Helminth Infection" instead of "Soil-Transmitted Helminths."

Thank you for sharing your feedback. We have revised the title to: “Health Workers' Perspectives on School-Based Mass Drug Administration Control Programs for Soil-Transmitted Helminthiasis and Schistosomiasis in Ogun State, Nigeria”. Also, in the study introduction the term Soil-Transmitted Helminths has been changed to “Soil-transmitted helminthiasis” (line 73)

2. Abstract: Specify the study duration, providing a specific timeframe for the intervention and assessments. Clearly articulate the research problem and objectives in the introduction. Provide a brief yet comprehensive background on the relevance and impact of soil-transmitted helminths (STH) infection and schistosomiasis. Establish a stronger connection between the study's objectives and the broader context of NTDs.

Thank you for your valuable input. The introduction has been modified in accordance to your recommendations (lines 24-33). Additionally, the study duration has been specified in line 36 (July to September 2022).

Abstract

Background: Soil-transmitted helminthiasis (STH) and schistosomiasis (SCH) are among the most prevalent neglected tropical diseases (NTDs), affecting 1.5 billion globally, with a significant burden in sub-Saharan Africa, particularly Nigeria. These diseases impair health and contribute to socio-economic challenges, especially in children, undermining educational and future economic prospects. The 2030 NTD Roadmap highlights Mass Drug Administration (MDA) as a critical strategy for controlling these NTDs, targeting vulnerable populations like school-age children. Despite some successes, challenges persist, indicating the need for deeper insights into program implementation. This study focuses on the perspectives of health workers implementing MDA in selected local government areas (LGAs) of Ogun State, Nigeria, aiming to identify challenges and enablers that align with the broader NTD 2030 goals.

Methodology/Principal Findings: The study used a qualitative research approach involving focus group discussions and in-depth interviews with health workers engaged in neglected tropical disease control programs in Ogun State, Nigeria, between July and September 2022. A semi-structured questionnaire guided the exploration of ideas, and the data were analyzed using the QRS Nvivo 12 software package. The study found that the school-based MDA control program's efficacy largely relies on strong collaborations and partnerships, particularly with educators, community heads, and other stakeholders. These alliances and strategic communication methods, like town announcements and media campaigns, have been pivotal in reaching communities. However, the program does grapple with hurdles such as parental misconceptions, limited funds, insufficient staffing, and misalignment with the Ministry of Education. It is recommended to boost funding, foster early stakeholder involvement, enhance mobilization techniques, and consider introducing a monitoring card system similar to immunization.”

3. Methodology/Principal Findings: Elaborate on the rationale for selecting the qualitative research approach. Clarify the criteria used for determining saturation in the absence of a formal sample size calculation. Elaborate on the process of pretesting interview topics and how reliability was ensured. Provide additional details on the content covered during the three-day training for data collectors. Consider providing more context on how discrepancies in coding were resolved among the research team. Clearly outline the selection criteria for health workers participating in the study. Provide additional details on the content of the semi-structured questionnaire. Include more information on the demographic characteristics of the study participants. Clarify the role of QRS NVivo 12 software in data analysis.

a) Elaborate on the rationale for selecting the qualitative research approach. 

Thank you for these comments. The rationale for the qualitative research approached has been stated in lines 124- 130. 

“A qualitative research approach was chosen for this study to delve deeply into the nuanced aspects of the MDA program. This approach was selected because it allows for a rich, detailed understanding of the complex realities and experiences of healthcare workers, educators, and other stakeholders involved in NTD control programs in Ogun State, Nigeria. Qualitative methods are particularly effective in capturing the diverse perspectives, attitudes, and social dynamics that quantitative methods might overlook, which is crucial for uncovering the underlying factors that influence the effectiveness of program implementation and addressing the barriers to success.”

b) Clarify the criteria used for determining saturation in the absence of a formal sample size calculation.

Thank you. We used three criteria to determine when we had reached saturation: repetition of key themes, emergence of a coherent pattern in the data, and judgment that additional interviews would likely not yield new insights. This has been extensively explained in lines 182-188.

c) Elaborate on the process of pretesting interview topics and how reliability was ensured

We appreciate this comment. This information has been added in lines 200-211.

“The pre-test of the interview topic involved a small group of individuals who had characteristics similar to the study population but were not part of the main study. Ensuring the reliability of the tool also encompassed refining the questions after pilot study to ensure clarity and the solicitation of the required information, validating findings with participants to ensure accurate reflection of their experiences and lastly engaging in peer debriefing”. 

d) Provide additional details on the content covered during the three-day training for data collectors. 

Thank you for your comment. The content covered during the three-day training for the interviewers has been provided in lines 213-221.

“The training introduced the research assistants to various qualitative data collection methods, such as IDIs, FGDs, and KIIs, each integral to comprehending the complex tapestry of human experiences. A significant portion of the training was devoted to ethical considerations, ensuring that the research assistants understood their role in conducting research with the utmost respect, dignity, and responsibility. Also, the training delved into ethnographic fieldwork, teaching research assistants to observe, participate, record, and reflect effectively. Role-playing exercises and field visit simulations were integral, sharpening their practical skills and preparing them for real-world complexities.”

e) Clearly outline the selection criteria for health workers participating in the study. 

Thank you. In lines 153-168, the new sentence reads; 

“Participants were carefully selected based on their roles within the health system, emphasizing direct responsibilities related to managing STH and SCH across different tiers of healthcare. The inclusion criteria focused on individuals actively involved in planning, executing, or evaluating control and prevention programs for NTDs in LGAs endemic with STH and SCH, representing a diverse mix of positions and expertise. The selection encompassed perspectives from policy and administration (DPH), local health programs (MoHs and LNTDs), and community interactions (CHEWs) to provide a comprehensive understanding of the challenges and strategies in managing NTDs, with attention to both experienced veterans and newer members.”

f) Provide additional details on the content of the semi-structured questionnaire. Include more information on the demographic characteristics of the study participants. 

Thank you for sharing your feedback. We have added a new table with the heading Key Elements of the interview guide in line 196.

Table 1

 Key Elements

FGDs 1 Community Illnesses & NTDs

 2 Prevalence and Vulnerable Groups

 3 Preventive Measures & Treatment

 4 Community Efforts & Drug Availability

 5 Program Awareness & Involvement

 6 Communication & Barriers

 7 Program Reception & Rumors

IDI/KIIs 1 Health Priorities & NTD Importance

 2 NTD Occurrence & MDA Program Awareness

 3 Communication & Barriers

 4 Program Reception & Rumors

 5 Data Management & Legislation

 6 Involvement & Health Sector Priorities

 7 Perceptions of NTD Programs

 8 Integration with Health Programs

 9 Program Evaluation & Stakeholder Involvement

 10 Policies & Suggestions

Also, more information on the demographic characteristics of the study participants has been stated in Table 3 (290).

Table 3: Socio-demographic characteristics of the study participants in the KIIs and FGDs.

Role Description Frequency

(N=41) Percentage (%)

CHEW Gender 

 Male 23 71.9

 Female 9 28.1

 Age in years 

 30-35 14 43.8

 36-40 5 15.6

 41-45 12 37.5

 46-50 1 3.1

 Marital Status 

 Single 4 12.5

 Married 21 65.6

 Divorced 5 15.6

 Widow 2 6.3

 Religion 

 Christianity 19 59.4

 Islam 13 40.6

 Years of experience 

 3- 5 12 37.5

 6-8 20 62.5

LNTD Gender 

 Female 4 100

 Age in years 

 36-40 1 25

 41-45 3 75

 Marital Status 

 Married 4 100

 Religion 

 Christianity 3 75

 Islam 1 25

 Years of experience 

 3- 5 2 50

 6-8 2 50

MOH 

 Gender 

 Male 2 50

 Female 2 50

 Age in years 

 41-45 1 25

 46-50 2 25

 51-55 1 50

 Marital Status 

 Married 3 75

 Divorced 1 25

 Religion 

 Christianity 2 50

 Islam 2 50

 Years of experience 

 9-12 1 25

 13-15 3 50

DoPH

 Gender 

 Male 1 100

 Age in years 

 46-50 1 100

 Marital Status 

 Married 1 100

 Religion 

 Christianity 1 100

 Years of experience 

 13-15 1 100

 Gender 

 Male 1 100

Clarify the role of QRS NVivo 12 software in data analysis

Thank you. The role has been clarified in lines 250-252.

“QRS NVivo 12 software played a pivotal role in the data analysis process of the study, serving as a tool for managing, organizing, and analyzing the unstructured qualitative data collected from FGDs and KIIs”

4. Results: Provide more context or examples to illustrate each thematic area in the results section. In other words, briefly explain the rationale for choosing the five thematic areas and their alignment with the research objectives. Provide additional information on the selection criteria for participants. Present participant demographics in more detail, such as years of experience, roles, etc. Define the demographic characteristics of participants before presenting thematic areas.

Thank you for this crucial observation. The rationale for the thematic areas has been stated as thus 

“to ensure an in-depth exploration of factors influencing the program efficacy and formulating informed recommendations” in lines 291-294.

For the selection criteria of the study participants the information has been presented in lines 153-168 as thus;

“Participants were carefully selected based on their roles within the health system, emphasizing direct responsibilities related to managing STH and SCH across different tiers of healthcare. The inclusion criteria focused on individuals actively involved in planning, executing, or evaluating control and prevention programs for NTDs in LGAs endemic with STH and SCH, representing a diverse mix of positions and expertise. The selection encompassed perspectives from policy and administration (DPH), local health programs (MoHs and LNTDs), and community interactions (CHEWs) to provide a comprehensive understanding of the challenges and strategies in managing NTDs, with attention to both experienced veterans and newer members”

The Table below has been inserted in the manuscript in line 290 to present the participants socio-demographic characteristics. 

“Table 3: Socio-demographic characteristics of the study participants in the KIIs and FGDs.

Role Description Frequency

(N=41) Percentage (%)

CHEW Gender 

 Male 23 71.9

 Female 9 28.1

 Age in years 

 30-35 14 43.8

 36-40 5 15.6

 41-45 12 37.5

 46-50 1 3.1

 Marital Status 

 Single 4 12.5

 Married 21 65.6

 Divorced 5 15.6

 Widow 2 6.3

 Religion 

 Christianity 19 59.4

 Islam 13 40.6

 Years of experience 

 3- 5 12 37.5

 6-8 20 62.5

LNTD Gender 

 Female 4 100

 Age in years 

 36-40 1 25

 41-45 3 75

 Marital Status 

 Married 4 100

 Religion 

 Christianity 3 75

 Islam 1 25

 Years of experience 

 3- 5 2 50

 6-8 2 50

MOH 

 Gender 

 Male 2 50

 Female 2 50

 Age in years 

 41-45 1 25

 46-50 2 25

 51-55 1 50

 Marital Status 

 Married 3 75

 Divorced 1 25

 Religion 

 Christianity 2 50

 Islam 2 50

 Years of experience 

 9-12 1 25

 13-15 3 50

DoPH

 Gender 

 Male 1 100

 Age in years 

 46-50 1 100

 Marital Status 

 Married 1 100

 Religion 

 Christianity 1 100

 Years of experience 

 13-15 1 100

 Gender 

 Male 1 100

In addition, the demographic characteristics of participants is now presented before the thematic areas (lines 285-288)

5. Discussion: Start the discussion with a concise recapitulation of key findings.

Explicitly connect health workers' knowledge and engagement to program success. Elaborate on potential challenges or limitations in forming and maintaining collaborations. Explore the role of technology or innovative communication methods in enhancing program effectiveness. Discuss the relative impact or severity of each identified barrier on program effectiveness. Discuss potential challenges or drawbacks of implementing the proposed recommendations. Consider prioritizing recommendations based on feasibility and impact. Emphasize the broader implications of the study findings for future research or policy changes.

Thank you for the essential information. The discussion section has been modified (lines 497 to 637”.)

“Discussion

Evaluating the understanding of health workers engaged in school-based MDA control programs regarding the challenges and facilitators of the program is crucial for its success (20) to ensure that comprehensive knowledge and necessary actions are aligned to achieve program objectives across all levels. According to Piotrowski et al. and Kabatereine et al., it's essential to consider insights from experienced health workers, the frontline individuals who directly interact with the target population in the program's initial stages (21, 22). Our study showed that health workers possess adequate knowledge of the program, demonstrating experience in their roles. Furthermore, it was evident that their engagement provided them with valuable opportunities for active community-based participation.

Collaboration is integral to the success of the MDA program (23). Project managers need to partner with stakeholders who can enhance the program's effectiveness. In this study, the collaborative efforts showcased by health workers, schools, and community leaders underscore the significance of partnerships in achieving success in MDA programs targeting STH and SCH. As highlighted by the CHEW, the involvement of teachers in the MDA program demonstrates a strategic approach to overcoming reluctance among children. This collaboration improves the acceptance of drugs and ensures that children receive essential treatment for these NTDs. Molyneaux et al. identified collaboration as essential for strengthening the health system. The NTD agenda extends beyond parasitic diseases, with a massive annual 

---

## [Editor Report · Decision Letter 1]

6 Feb 2024

PONE-D-23-31424R1Health Workers' Perspectives on School-Based Mass Drug Administration Control Programs for Soil-Transmitted Helminthiasis and Schistosomiasis in Ogun State, Nigeria.PLOS ONE

Dear Dr. Akinsolu,

Thank you for submitting your manuscript to PLOS ONE. After careful consideration, we feel that it has merit but does not fully meet PLOS ONE’s publication criteria as it currently stands. Therefore, we invite you to submit a revised version of the manuscript that addresses the additional points raised by the Academic Editor during the review process. Your revised submission would be shared with original reviewers for additional comments.

We look forward to receiving your revised manuscript.

Kind regards,

Hammed Oladeji Mogaji, B.Sc. M.Sc Ph.D

Academic Editor

PLOS ONE

Journal Requirements:

Additional Editor Comments:

Editors comments

Foremost, in line with PLOS ONE recommendations, authors are advised to use the COREQ checklist, or other relevant checklists listed by the Equator Network, such as the SRQR, to ensure complete reporting (http://journals.plos.org/plosone/s/submission-guidelines#loc-qualitative-research). In general, we would expect qualitative studies to include the following: 1) defined objectives or research questions;

2) description of the sampling strategy, including rationale for the recruitment method, participant inclusion/exclusion criteria and the number of participants recruited;

3) detailed reporting of the data collection procedures;

4) data analysis procedures described in sufficient detail to enable replication;

5) a discussion of potential sources of bias;

and 6) a discussion of limitations.

Another general comment: most of the responses of the authors to the reviewers should have made it to the text, but rather could have been an explanation or response back to the reviewers; an example could be found under the sub-sections study design, sample size determination; and a couple of others. Authors need to consider cleaning the manuscript up in the next round, by making inputs that goes into the text more succinct.

Most references cited in the first paragraph of the introduction, are not appropriate and authoritative for the associated quotes. Same applies to Reference 9. Authors should recheck. Citation should be from WHO, Hotez and a couple of others who gave this foundational metrics.

Paragraph 2. Same applies to Reference 9. Also, the definition of MDA here should be clearer than this; the concept of delivery of at risk population without prior diagnosis is very important, same as the need to mention the medicines used for the diseases, same as the strategy of using drug distributors, also the recommended guidelines based on endemicity, and the objective of the entire process, which is to meet 75% therapeutic coverage to interrupt transmission cycle.

Towards the later end of paragraph 2. Authors only cited Ref 15. Since they mentioned “certain studies”. They infact need to futher strengthen this claim with a couple more references. If they have local programmatic data/evidence in the study area they have chosen that would also be more relevant.

Paragraph 3. Authors could do more by briefly highlighting the roles of health workers in the entire MDA process. They only mentioned health workers play a crucial role, without highlighting those roles. This would be useful for readers not familiar with MDA process.

Paragraph on study settings; the % reported here and the inconsistent support for partner programs are not referenced. Also, the ref 18 is old, as there are more recent estimates on schistosomiasis prevalence in the study location; you can revise by complementing with newer references.

Study design:

Authors should briefly describe their design here; and follow the suggestion below;

Can you take these lines to the introductory section, to complement why you have chosen a qualitative approach

‘A qualitative research approach was chosen for this study to delve deeply into the nuanced aspects of the MDA program. This approach was selected because it allows for a rich, detailed understanding of the complex realities and experiences of healthcare workers, educators, and other stakeholders involved in NTD control programs in Ogun State, Nigeria. Qualitative methods are particularly effective in capturing the diverse perspectives, attitudes, and social dynamics that quantitative methods might overlook, which is crucial for uncovering the underlying factors that influence the effectiveness of program implementation and addressing the barriers to success.”

Can you take these lines to the data analysis section, to complement why you have employed a thematic analysis?

“Thematic analysis was employed to extract key themes from the data collected. This qualitative technique was instrumental in providing a comprehensive understanding of the challenges faced by NTD control programs in addressing STH and SCH. It offered valuable insights into the barriers to achieving effective control interventions and highlighted the importance of understanding the context-specific factors that impact the success of health initiatives”

Sampling technique:

Can the authors provide a flow chart on how they performed these selections, since several criteria and steps were involved in the selection of these stakeholders. Your flow chart can also show the number of stakeholders available at each level, and how you have sampled them (selected a few of them) etc

Sampling technique:

Please remove this sentence

“QRS NVivo 12 software played a pivotal role in the data analysis process of the study, serving as a tool for managing, organizing, and analyzing the unstructured qualitative data collected from FGDs and KIIs.”

Results

For your tables, please provide a footnote describing, what CHEW, LNTD, MOH, DoPH means

You can re-draw your table 3 without having so much rows, but instead columns, merge frequency and % in a column, and also, make the Roles as column headers;

---

## [Author Response · Author response to Decision Letter 1]

17 Mar 2024

Dear Editor,

We are reaching out to present the revision of the manuscript that my colleagues and I have diligently worked on, titled "Health Workers' Perspectives on School-Based Mass Drug Administration Control Programs for Soil-Transmitted Helminths and Schistosomiasis in Ogun State, Nigeria" with manuscript ID PONE-D-23-31424R1. 

We have carefully considered each comment and have made corresponding revisions to the manuscript. Also, we have tried to ensure that the methodology adheres to the COREQ checklist.

Below, we address each comment in detail and outline the changes we have made to improve our work.

Specific Comments:

1. Most references cited in the first paragraph of the introduction, are not appropriate and authoritative for the associated quotes. Same applies to Reference 9. Authors should recheck. Citation should be from WHO, Hotez and a couple of others who gave this foundational metrics. 

Thank you for your recommendation. The references have been updated appropriately as suggested.

2. Also, the definition of MDA here should be clearer than this; the concept of delivery of at risk population without prior diagnosis is very important, same as the need to mention the medicines used for the diseases, same as the strategy of using drug distributors, also the recommended guidelines based on endemicity, and the objective of the entire process, which is to meet 75% therapeutic coverage to interrupt transmission cycle.

Thank you for your observation. The definition of MDA has been modified; it is written as thus. 

“The 2030 NTD Roadmap has a strategic focus on controlling and eliminating NTDs, which has a significant portion of its success in implementing integrated mass drug administration (MDA) programs [9, 10] as the primary approach for controlling and potentially eliminating NTDs. MDA involves periodically administering safe, effective, low-cost drugs to at-risk populations until disease-specific targets are met [11].” (See lines 80-84)

3. Same applies to Reference 9. Authors should recheck. Citation should be from WHO, Hotez and a couple of others who gave these foundational metrics.

Thank you. The sentence has been supported with other references as recommended.

4. Authors only cited Ref 15. Since they mentioned “certain studies”. They in fact need to further strengthen this claim with a couple more references. If they have local programmatic data/evidence in the study area they have chosen that would also be more relevant.

Thank you for your observation. We added the following sentences to more explanation in the paragraph. 

“As recommended by the WHO, a single dose of albendazole (400 mg) or mebendazole (500 mg) in MDA to treat STH is efficient [10, 14]” (See lines 88-89)

In line 93

“Notable, the MDA program initiative has played a pivotal role in alleviating the burden of SCH and STH [15, 16]. In an endemic area, the WHO recommends that at least 75% of school-age children (SAC) should receive MDA [17, 18]” (See lines 89-91)

Additionally, more references have been added to strengthen our claim in the statement.

Griswold E, Eigege A, Adelamo S, Mancha B, Kenrick N, Sambo Y, et al. Impact of three to five rounds of mass drug administration on schistosomiasis and soil-transmitted helminths in school-aged children in north-central Nigeria. J The American Journal of Tropical Medicine Hygiene. 2022;107(1):132. 

Agboraw, E., Sosu, F., Dean, L. et al. Factors influencing mass drug administration adherence and community drug distributor opportunity costs in Liberia: a mixed-methods approach. Parasites Vectors 14, 557 (2021). https://doi.org/10.1186/s13071-021-05058-w

Makaula, P., Kayuni, S. A., Mamba, K. C., Bongololo, G., Funsanani, M., Musaya, J., Juziwelo, L. T., & Furu, P. (2022). An assessment of implementation and effectiveness of mass drug administration for prevention and control of schistosomiasis and soil-transmitted helminths in selected southern Malawi districts. BMC health services research, 22(1), 517. https://doi.org/10.1186/s12913-022-07925-3

Asfaw MA, Hailu C, Beyene TJ. Evaluating Equity and Coverage in Mass Drug Administration for Soil-Transmitted Helminth Infections among School-Age Children in the Hard-to-Reach Setting of Southern Ethiopia. Pediatric Health Med Ther. 2021 Jul 8;12:325-333. doi: 10.2147/PHMT.S316194. PMID: 34267576; PMCID: PMC8275865.

World Health Organization. Guideline: Preventive Chemotherapy to Control Soil-Transmitted Helminth Infections in At-Risk Population Groups. World Health Organization; 2017. [PubMed] [Google Scholar]

5. Authors could do more by briefly highlighting the roles of health workers in the entire MDA process. They only mentioned health workers play a crucial role, without highlighting those roles. This would be useful for readers not familiar with MDA process.

Thank you for your valuable input. We have added the roles of health workers.

“Health workers facilitate MDA by distributing medication, educating communities, and monitoring progress to control and eliminate diseases effectively. Therefore, the health worker’s perceptions will provide insights into the challenges and enablers of the MDA control programs and significantly improve the programs, thus aiding in achieving the NTD 2030 goals [21-23]” (See lines 99-103)

6. Can the authors provide a flow chart on how they performed these selections, since several criteria and steps were involved in the selection of these stakeholders? Your flow chart can also show the number of stakeholders available at each level, and how you have sampled them (selected a few of them) etc.

Thank you for your recommendation. The flow chart below showing the number of stakeholders and the number sampled has been added as Figure 1 (Flow Chart of Stakeholder’s Selection) (See line 157).

7. You can re-draw your table 3 without having so much rows, but instead columns, merge frequency and % in a column, and also, make the Roles as column headers.

Thank you for your feedback. The table has been modified, See line 131. 

We have attached the revised manuscript for your perusal. We understand the review process is meticulous and time-consuming, so we sincerely appreciate your attention to our work. We're eagerly looking forward to your invaluable feedback.

---

## [Decision Letter · Decision Letter 2]

2 Apr 2024

PONE-D-23-31424R2Health Workers' Perspectives on School-Based Mass Drug Administration Control Programs for Soil-Transmitted Helminthiasis and Schistosomiasis in Ogun State, Nigeria.PLOS ONE

Dear Dr. Akinsolu,

Thank you for submitting your manuscript to PLOS ONE. After careful consideration, we feel that it has merit but does not fully meet PLOS ONE’s publication criteria as it currently stands. Therefore, we invite you to submit a revised version of the manuscript that addresses the points raised during the review process.

We look forward to receiving your revised manuscript.

Kind regards,

Hammed Oladeji Mogaji, Ph.D

Academic Editor

PLOS ONE

Journal Requirements:

Reviewers' comments:

Reviewer's Responses to Questions

**Comments to the Author**

1. If the authors have adequately addressed your comments raised in a previous round of review and you feel that this manuscript is now acceptable for publication, you may indicate that here to bypass the “Comments to the Author” section, enter your conflict of interest statement in the “Confidential to Editor” section, and submit your "Accept" recommendation.

Reviewer #2: (No Response)

2. Is the manuscript technically sound, and do the data support the conclusions?

Reviewer #2: Yes

3. Has the statistical analysis been performed appropriately and rigorously? 

Reviewer #2: Yes

4. Have the authors made all data underlying the findings in their manuscript fully available?

Reviewer #2: (No Response)

5. Is the manuscript presented in an intelligible fashion and written in standard English?

Reviewer #2: Yes

6. Review Comments to the Author

Reviewer #2: (No Response)

7. PLOS authors have the option to publish the peer review history of their article (what does this mean?). If published, this will include your full peer review and any attached files.

Reviewer #2: **Yes: **Dr Ogechukwu B. Aribodor

---

## [Author Response · Author response to Decision Letter 2]

5 Apr 2024

Response to Reviewer's Comments:

We would like to express our gratitude to the reviewer for the constructive feedback on our manuscript. We appreciate the acknowledgment of our efforts in improving the clarity and engagement of the content. We have carefully considered the comments provided and addressed them accordingly:

Author Summary: We acknowledge the reviewer's comment regarding the clarity of the sentence regarding the strategies employed by health workers to reach out to communities. We have rephrased the sentence to enhance clarity and ensure that the message is effectively conveyed to readers. The statement now reads:

“The health workers said that teaming up with teachers and community leaders, and using different ways to talk to people, has made it easier to connect with communities.”

Introduction: We thank the reviewer for suggesting the inclusion of the commencement date of the Mass Drug Administration programs in Ogun State to provide better context for readers. We have incorporated the information into the study settings of the methodology. 

Ogun State, Nigeria, was selected as the study location due to its high prevalence of STH (19.2%) and SCH (32.2%) and the inconsistent support of partner programs for NTD control [24, 25]. MDA commenced in Ogun State 22 years ago directed to STH and SCH treatment [25].

Results: We recognize the importance of including footnotes for abbreviations such as "NA" in the tables to ensure clarity for readers. 

We have added footnotes to Table 4.

“NA: Not Applicable”

Acknowledgment Section: We appreciate the reviewer for bringing attention to the inconsistency between the acknowledgment statement and the information provided in the Financial Disclosure Section regarding the funding of the study. 

We have reviewed the funding sources of the research.

We are also grateful to the Department of Public Health, Faculty of Basic Medical and Health Sciences, Lead City University, Ibadan, Nigeria; Nigerian Institute of Medical Research, Lagos State; and the University of Debrecen, Hungary, for providing the necessary facilities and support for this research.

Once again, we thank the reviewer for their valuable feedback, which will undoubtedly contribute to the overall quality and clarity of our manuscript.

---

## [Editor Report · Decision Letter 3]

8 Apr 2024

Health Workers' Perspectives on School-Based Mass Drug Administration Control Programs for Soil-Transmitted Helminthiasis and Schistosomiasis in Ogun State, Nigeria.

PONE-D-23-31424R3

Dear Dr. Akinsolu,

We’re pleased to inform you that your manuscript has been judged scientifically suitable for publication and will be formally accepted for publication once it meets all outstanding technical requirements.

Kind regards,

Hammed Oladeji Mogaji, Ph.D

Academic Editor

PLOS ONE